**Data Availability Statement:** All relevant data are within the manuscript and its Supporting Information files.

# Informed consent in cancer clinical care: Perspectives of healthcare professionals on information disclosure at a tertiary institution in Uganda

**Rebecca Kampi[1], Clement Okello[2], Joseph Ochieng[1], Erisa Sabakaki Mwaka🔟[1] ***

1 School of Biomedical Sciences, College of Health Sciences, Makerere University, Kampala, Uganda,
2 Uganda Cancer Institute, Kampala, Uganda

* erisamwaka@gmail.com

## Abstract

### Introduction

While there have been several studies examining the understanding and quality of informed consent in clinical trials of cancer therapies, there is limited empirical research on health practitioners' experiences on the informed consent process in cancer care, especially from low resource settings. This study explored health professionals' perspectives on information disclosure during the consenting process in cancer care.

### Methods

A qualitative descriptive approach was used to collect data. Face to face interviews were conducted with 10 purposively selected healthcare professionals who were actively involved in soliciting informed consent at a cancer treatment centre in Uganda. A thematic approach was used to interpret the results.

### Results

There were five key themes, and these included information disclosure to patients; assessment of patients' cancer awareness, treatment preferences and expectations; informed consent practices; barriers to optimal informed consent and information disclosure; and recommendations for improving the consenting process. All respondents appreciated the value of disclosing accurate information to patients to facilitate informed decision making. However, the informed consent process was deemed sub-optimal. Respondents asserted that patients should be the psychological wellbeing of patients should be protected by mentally preparing them before disclosing potentially distressing information. All healthcare professionals were appreciative of the central role the family plays in the consenting process.

### Conclusion

Overall, informed consent practices were not ideal because of the several challenges. Inadequate time is devoted to information disclosure and patient education; there is lack of

**Funding:** This study was supported by the Fogarty International Center of the National Institutes of Health under Award Number R25TW009730 that supported RK's Master of Health Sciences in Bioethics program at Makerere University. The content is solely the responsibility of the authors and does not necessarily represent the official views of the National Institutes of Health. The authors would like to acknowledge all respondents. The funders had no role in study design, data collection and analysis, decision to publish, or preparation of the manuscript.

**Competing interests:** No authors have competing interests.

privacy; and informed consent documentation is poor. There is a need for significant improvement in informed consent practices and healthcare professional-patient communication.

## Introduction

Informed consent is an ethical imperative before any medical procedure. Informed consent is premised in the ethical principle of autonomy that acknowledges a person's right to hold views, to make choices, and to take actions based on his/her values and beliefs [1, 2]. Beauchamp and Childress suggested two different senses of informed consent. In the first sense, they define informed consent as "an individual's autonomous authorisation of a medical intervention or of participation in research" [1]. In this sense, informed consent is considered to be an "autonomous authorisation" when a patient intentionally, voluntarily and with sufficient understanding authorises a physician to act. Autonomous authorisation is in direct contrast to the paternalistic approach of physicians who make independently conceived treatment decisions in the best interest of the patient [3]. In the second sense, informed consent refers to "conformity to the social rules of consent that require professionals to obtain legally or institutionally valid consent from patients or subjects before proceeding with diagnostic, therapeutic, or research procedures" [1]. In this case, physicians who obtain consent as an institutional obligation can and often fail to meet the ethical standard of the autonomy-based model. Autonomy-protecting rules may be difficult to implement. Therefore, physicians should not only evaluate institutional rules in terms of respect for autonomy and autonomous authorisation, but also in terms of the potential repercussions of imposing stringent requirements on institutions and professionals [1].

In cancer therapy, healthcare professionals should aim at a satisfactory informed consent process since care treatments profoundly impact on a patient's health and quality of life [1, 4]. However, there are several factors that influence the consenting process in cancer care. Fluctuations in a patient's clinical condition, symptom control, and receipt of prognostic information may affect a patient's motivation to contribute to consent discussions [5]. Furthermore, people with cancer may experience intense emotional distress, bewilderment, and are often challenged with symptomatic disease and closeness to death [5]. This circumstance significantly impacts their support needs and personal relationships, psychological wellbeing, and presents challenges for patients and families during the consenting process. In some cultures, unpleasant facts about the prognosis of disease conditions are unwelcome and are better withheld [6–8]. Some patients may feel discomfort talking about death and dying, may have strong emotional responses, and others may be in denial of the diagnosis and prognosis [9]. Religion and spirituality have been reported to have positive impact on cancer care [10–12]. However, while religion may be a source of comfort and spiritual relief, it may also be used to support denial about cancer progression or support futile medical care at the end of life. These socio-cultural factors may impede a patient's ability to engage in conversations and ability to ask relevant questions during the consenting process [9].

While there have been several studies examining the understanding and quality of informed consent in clinical trials of cancer therapies [13–17], there is limited empirical research on health practitioners' experiences on the informed consent process in cancer care, especially from low resource settings. To the best of the authors' knowledge, this is the first article to document informed consent practices in cancer care in Uganda. In Uganda, 34,008 new cases and

22,992 deaths due to cancer were registered in 2020 [18]; and there is only one health facility in the country that provides comprehensive cancer centre to a population of 45.7 millions [18]. It is also important to note that there is no organized national cancer screening program, and the cancer referral system is sub-optimal; as such, many patients are never referred and may never seek treatment [19, 20]. Cancer care is often delayed due to challenges such as lack of knowledge and awareness of clinical presentation of cancer, poor access to diagnostic and therapeutic services, stigma and financial constraints [21, 22]. Nakaganda et al. in a study that assessed the challenges faced by cancer patients in Uganda reported that 55% participants had limited knowledge about their disease and treatment, 41% were never counselled about the side effects of therapy, and 38% were not given instructions before initiating treatment [21]. It is evident from the above discussion that there is a glaring gap in the provision of health information to cancer patients. Furthermore, several studies have reported sub-optimal informed consent practices in clinical care in Uganda [23–27]. Given the sensitive nature of the information provided to cancer patients, particularly regarding prognosis, treatment options, the risks and effects of treatment, it is important to understand the perspective of health professionals on the consenting process for cancer care as a basis for further optimisation of the informed consent. This study therefore set out to explore the perspectives of healthcare professional on information disclosure during the informed consent process for cancer care.

## Materials and methods

A qualitative descriptive approach was used to collect data on healthcare professionals' experiences and perspectives on information disclosure during the consenting process for cancer clinical care at a Cancer Treatment Centre in Uganda, that will hereafter be referred to as The Centre. A qualitative descriptive approach provides a comprehensive summary and straight forward description of the phenomenon [28]. The Centre has several wards and specialized out-patient clinics. The Centre also has three regional service centres and offers specialized cancer care services. The Centre is a centre of excellence that with a mandate for cancer prevention and control, specialized care, research, and training. The Centre offers out-patient, in-patient, and outreach services. It also offers palliative and rehabilitation services. The Centre also has a comprehensive community cancer program that takes lead in cancer prevention through screening, health education, prevention, early detection, and access to the right treatment.

### Participants

Participants included purposively selected healthcare professionals based on their role in obtaining informed consent for clinical care in the out-patient clinics at The Centre. All participants routinely participated in obtaining informed consent for clinical care in the out-patient clinics at The Centre. This study focused on healthcare professionals because several studies have reported challenges to the informed consent process in clinical care in Uganda, with healthcare professionals being the main culprits [23–27].

### Data collection

Ten key informant interviews were conducted between September 2019 to May 2020 by a team of three researchers that included a bioethicist (ESM), a master of bioethics graduate student (RK) and a physician (CO). Prospective participants were physically approached, given a brief description of the study and a convenient appointment made. All the individuals approached participated in the study. Face to face Interviews were conducted in a private quiet room at The Centre after obtaining written informed consent. Prior to commencing the study,

the research team was trained on the protocol to ensure that they internalized and understood the study well. Data were collected using a semi-structured key informant interview guide that was developed by ESM and RK, and consisted of open-ended questions on what information is disclosed to patients; how the information is disclosed; informed consent practices; challenges to the consenting process; and recommendations for improving the informed consent process for cancer care. The interview guide was pre-tested on three healthcare professional at another hospital and revised prior to the full data collection process. A team of three researchers (RK, ESM and CO), conducted all interviews to ensure consistency. All interviews were conducted in English, audio recorded with permission from respondents alongside detailed note taking, and later transcribed verbatim. Preliminary data analysis was performed after five interviews and the codes identified. Analysis was repeated after the eighth interview and there were no new codes emerging from the interviews. We conducted two more interviews to ensure that data saturation had been achieve [29]. Transcripts were cleaned before the analytical process. On average, interviews lasted between 45–60 minutes. Debriefing meetings were held by the research team at the end of each interview to ensure completeness and to also review preliminary perspectives that had arisen.

## Data management and analysis

Data analysis was conducted continuously throughout the study using a thematic approach [30, 31]. Two of the authors (RK and ESM) then developed a codebook and coding framework. Themes were then generated both deductively, based on our prior analytic framework derived from the literature on informed consent for clinical care, as represented in the interview guide (see S2 File); and inductively, by considering the new themes that emerged from the text. Then RK, ESM, CI and JO examined the themes for patterns until consensus was achieved on the final themes after several iterative discussions. Codes and themes were organized using NVivo 12 software (QSR International Pty Ltd, 2014). The authors identified quotes in the respondents' transcripts that were congruent with the overarching themes. Three transcripts were returned to interviewees to check for accuracy of transcription [32]. The authors had a well-established relationship. RK is a female and has a Master of Health Science in Bioethics but was a graduate student at the time of the study. This manuscript is part of her master's dissertation. ESM (Master of Social Science in Health Research Ethics, PhD) and JO (Master of Health Sciences in Bioethics) are male practicing bioethicists and faculty members of the International Health Research Ethics Training program at Makerere University; and CO (Master of Medicine) is a male haematologist. ESM, and JO were mentors and academic supervisors to RK.

## Ethical considerations

Ethics approval was obtained from the Makerere University School of Biomedical Sciences Higher Degrees and Research Ethics Committee (SBSHDREC-675). This study was conducted according to the principles expressed in the Declaration of Helsinki (2013). Written informed consent was obtained from all participants prior to the interviews. Data was kept securely, and all recordings and transcripts were de-identified, assigned special codes and stored on a password-protected computer. No participant identifying information was published.

## Results

Ten healthcare professionals participated in this study as shown in Table 1. They included four doctors, four nurses, a social worker, and a health educator.

Findings are summarized under five themes (Table 2).

**Table 1. Demographic characteristics.**

| Sex | Male | 5 |
|---|---|---|
| | Female | 5 |
| **Occupation** | Nurse | 4 |
| | Doctor | 4 |
| | Social worker | 1 |
| | Health educator | 1 |

## Information disclosure to patients

Information disclosure and comprehension is core to valid informed consent; and this is critical when dealing with patients with life-threatening disease conditions like cancer.

Most respondents appreciated the fact that patients have a right to information relevant to their disease. They indicated that disclosure of adequate relevant information facilitates proper decision making in medical care.

*It can help them to make good decisions. In some instances when you lack specific services. . .* [It] *helps when they have the capacity to go to another hospital or outside like if someone wants a bone marrow transplant [. . .]* (UCI 08).

The use of decisional aids was deemed important in enhancing patient understanding. Respondents said that they use various information, education, and communication (IEC) materials to deliver information to patients including flip charts, cancer specific booklets and information leaflets, illustrative diagrams, and several others. However, one nurse pointed out that IEC materials are insufficient and healthcare professionals seldom utilize them.

*Patients' information on cancer or taking care of a cancer patient is available. They may be in books, some are in leaflets, but they are very inadequate, and they are not frequently being used. (UCI 04)*

Some respondents mentioned that at times they withhold or delay disclosure of vital information to patients they feel are mentally unstable or may not be in position to understand. They said that occasionally, withholding information is done at the behest of patients' family members. Withholding information was reportedly aimed at protecting patients' psychosocial wellbeing, give them hope, and encourage adherence to treatment.

*There are health professionals that believe that if you give information, you destroy the patients' hope so they stop you and say you [*may] *discourage our patients. (UCI 10)*

Another respondent noted that occasionally doctors use deception when managing cancer patients to protect their wellbeing and ensure adherence to treatment.

*I have experienced some doctors who are trapped between saying the truth and lying to the patient . . .because this doctor now doesn't want to tell this patient that my dear this will not help you [. . .] Lies are happening.* (UCI 09)

On the other hand, two respondents emphasized the need for veracity. They pointed out that truth telling was key to building trust and confidence in the medical team. They also emphasized the importance of assessing a patient's readiness to receive bad news. They

**Table 2. Themes and key findings.**

| Themes | Key findings | Quotes |
|---|---|---|
| 1. Information disclosure to patients | • Patients have a right to information.<br>• Adequate information facilitates informed decision making<br>• Use of decisional aids<br>• Withholding of information<br>• The need for veracity<br>• Breaking of bad news<br>• Delegation of informed consent administration | *We try as much as possible to open up and also tell them what we expect them to do, one of the things we tell them is we really want to do the investigations and we tell them the plan of how they should handle their treatment when they get to initially start on it. We tell them about the frequency of the routine clinic appointments [. . .] (UCI 04)*<br>*When you have a very terminally ill patient, or very advanced and you gauge that his or her mental health is not adequate to take in this information [. . .] or giving the information could be more harmful than of benefit to the patient, sometimes you may defer that information. First manage some symptoms when stronger; we bring out the topic on prognosis. But the disadvantage also is that there is delay in taking critical decisions (UCI 06).*<br>*Yeah, just understanding the principles of breaking bad news you have to know how much information you give at a go. . . we know that if you give them too much information, they will only hear the diagnosis and they will not comprehend to the rest (UCI 10).* |
| 2. Assessment of patients' cancer awareness, treatment preferences and expectations | • Assessment of prior knowledge<br>• Patients' expectations and preferences | *At the first encounter we want to find out how much that person knows about the condition to guide our conversation. . . and then what are their expectations from care. Once we know that then we can know how to communicate further (UCI 10).* |
| 3. Informed consent practices | • Disclosure of intricate information should be done by doctors.<br>• Poor informed consent documentation<br>• Involvement of patients and their families in decision making | *Even myself I may not know which procedure. . . so what is asked from patients is verbal consent. . .and many times its probably more of one sided like; we are going to do this procedure [. . .] (UCI-10).*<br>*Normally this informed consent that is obtained is not cancer specific it is for general-purpose treatment. When opening a file, there is a general sheet that is not cancer specific. . . . (UCI 06).*<br>*We do family conferences in cancer treatment especially when we realize that the patient is having challenges [. . .] We prefer to bring a bigger family and try to talk about the same thing and may be trying to find out why the patient is declining in receiving whatever we give the patient. (UCI 05)* |
| 4. Barriers to optimal informed consent and information disclosure | • Patients are not given enough time to ask questions and understand<br>• Limited knowledge of healthcare providers on cancer<br>• Bad attitude and poor communication skills of some healthcare professionals<br>• Language barriers<br>• Lack of privacy | *Time is a challenge because you're meant to give a patient time to make an informed decision, they don't even have to sign on that day. They can even go and come back on another day but here it is the reverse because you open the file on that day so they make an informed decision, so I think it's mainly time (UCI 05).*<br>*Giving information needs time, the time the doctor spends with the patient is so short that patients cannot understand. Communication needs time but the time we have to communicate with patients is minimal (UCI 09).*<br>*There are no policy guidelines on how to give informed consent to cancer patients, when, by who and to who as well as its implication (UCI 09).*<br>*Especially if they communicate a preference which is against the health care providers decision for example, when they say I don't want radiotherapy then the person is offended and says you go home. So sometimes it happens (UCI 10).* |
| 5. Recommendations for improving the consenting process for cancer care. | • Training of healthcare professionals and continuing medical education<br>• Developing of appropriate and detailed informed consent documents, and standard operating procedures<br>• Improvement of physical infrastructure<br>• Involvement of patients in decision-making | *[. . .] the personnel need to have training" (UCI 02)*<br>*I think there must be a little revision on the policy or guidelines of consent because its' like we are using the 1900 agreement, things have changed a lot. There must be a policy which caters for the participation of the clients in the informed consent [. . .] need to study the practicability of the current consent system and whether it meets the criteria of consent. (UCI 09)*<br>*They have to improve on materials and have to provide facilities or rooms where you can have discussions like that like here ideally, we need to have some chairs here, more space (UCI 08).* |

contended that patients should first be mentally prepared before communicating potentially distressing information, as stated in this quote.

> *I think it's not ideal for every patient to get their information when mentally not stable [. . .] because there are instances where people are given information and that's when they stop talking and they die. (UCI 08)*

Assessment of patients' cancer awareness, treatment preferences and expectations

It is important to assess patients' awareness, prior knowledge, and perceptions of cancer during the informed consent process. Respondents indicated that they try to obtain as much baseline information from patients as possible.

> *. . .. we want to find out how much that person knows about the condition [. . .] this guides our conversation and the information we give subsequently. How much do they know about the condition and then what are their expectations from care? Once we know that then we can know how to communicate further. (UCI 10)*

Doctors reported that they inquire about the patient's health condition, past medical and treatment history, beliefs and perceptions of cancer, and treatment expectations. However, some respondents noted that by the time patients present to The Centre, they have already received medical care from several service providers including traditional healers, who unfortunately misguide them. Patients were also reported to receive a lot of conflicting information from the community that ends up confusing them.

> *90% of the cancer patients who come here have been elsewhere and that has implications. The first one is financial because they have spent a lot of money. Secondly, they have a lot of information which they have been given. Thirdly, patients have been given different concoctions as well as trying to get remedies either to their pain or swelling or to whatever problem they have. So, I think it is very important when you come, we try to know your beliefs. (UCI 05)*

## Informed consent practices

Overall, all respondents admitted that the consenting practices at The Centre were not ideal. They noted that whereas disclosure of intricate information and treatment planning is supposed to be done by "senior doctors", this responsibility is usually delegated to junior doctors, nurses, counsellors, and the palliative care team. In some cases, even record clerks participate in soliciting consent.

> *The records personnel, they are the ones who make them sign. . .But I don't know how they delegate duties. (UCI 02)*

Most respondents expressed dissatisfaction with the way informed consent is documented at The Centre. They reported that the informed consent forms used, particularly for admission and clinical care, are deficient, not specific to cancer and are devoid of detail. They noted that patients are requested to sign one general consent form at admission irrespective of the procedures to be undertaken.

> *I don't know if they really understand but I think the blanket informed consent that we use is not enough because we make them sign in the beginning that they will accept all the treatment given and all the procedures and that is it but even it's a blanket consent. (UCI 10)*

It was also established that patients just give *verbal consent* for medical procedures, some of which are complex and invasive.

Respondents pointed out that patients are usually not given the chance to voice their opinions or ask questions, they just must accept what the clinician decides for them. However, a doctor noted that patients are increasingly advocating for more active involvement in decision-making for issues concerning their medical care.

*People's autonomy is very dominant in ethics, but things have changed [. . .] In the past there was this holistic approach that would be sort of paternalistic, but patients started arguing. How do you know that it is the best for me? I have a right to make my decision to determine what is best for me. (UCI 10)*

Respondents acknowledged the need to involve patients in meaningful dialogue during the consenting process, and the need to consider patient preferences. However, they noted that oftentimes accommodating patients' preferences is difficult.

*I think there is room for negotiation. However, the challenge with cancer treatment is for each cancer you have a protocol. So, the negotiation may be that you have breast cancer if you have stage one and stage 2 the protocol is ABCD. Do you agree to follow ABCD? Then I will tell you depending on [the patient's] response, if we follow ABCD our chances are going to be better . . .. but it is your choice if you want to receive treatment it solely depends on you. (UCI 05)*

All respondents appreciated the role of the family in the care of people living with cancer. Three respondents indicated that it is important to involve family members in the decision-making process because caring for cancer patients requires a multidisciplinary approach. A doctor reported that family conferences are usually convened to make important decisions regarding patients' treatment, especially for the very sick and the terminally ill. Another doctor however noted that at times, the treatment preferences of the family may diverge from the patient's wishes.

*Family resistance is a big factor which may interfere with patients' treatment. They want to know, like they want to override the patient [. . .] In that case we normally take the patients decision* [. . .] (UCI 06)

All doctors reported that there were major weaknesses in the documentation of informed consent. They complained about the poorly designed informed consent forms.

*I think maybe the limitations are that maybe we don't have* [consent] *forms where patients can sign for particular procedures [. . .] so probably we do blanket consent. (UCI 10)*

### Barriers to optimal information disclosure during the informed consent process

Several barriers to optimal informed consent were identified. Most respondent conceded that they had challenges with information disclosure because of the heavy patient load, limited space, lack of manpower and lack of time. All doctors agreed that patients are not given adequate time, as such, information disclosure is at times inadequate.

*Adequacy of information is a limitation on our part. We are not able to go through the full depth of the information due to big patient numbers [. . .] and we feel the clinicians are not enough. The number of cancer patient cases have gone up to about three times while the*

*number of health workers has also gone up, but not as high as the patient number so there are still gaps. (UCI 06)*

*Giving information needs time, the time the doctor spends with the patient is so short that patients cannot understand. Communication needs time but the time we have to communicate with patients is minimal. (UCI 09)*

Respondents observed that some of the healthcare professionals that obtain informed consent have insufficient knowledge on cancer, so they are not in position to offer detailed explanation to patients. They attributed this to poor training, allocating roles to unqualified peoples and the lack of standard operating procedures for obtaining informed consent.

*There are no policy guidelines on how to give informed consent to cancer patients, when, by who and to who as well as its implication. (UCI 09)*

Another perceived barrier to optimal informed consent is the bad attitude of some healthcare professionals. Respondents reported that some healthcare providers are rude and unfriendly, as such, patients are usually afraid of opening up, and asking questions. One respondent reported that patients are repulsed by this negative attitude.

*I have heard about patients who are like; oh no! I don't want to be seen by that specific person he doesn't explain well for me to understand* [. . .] (*UCI 05*)

Furthermore, some healthcare professionals reportedly do not respect patients' preferences; they take offence when patients make suggestions that are contrary to their position.

*Especially if they communicate a preference which is against the healthcare providers decision. For example, when they say I don't want radiotherapy then the person is offended and says you go home, so sometimes it happens.* (UCI 10)

The Centre receives patients of various ethnicities, and this causes linguistic challenges. Respondents reported that they usually find problems communicating with patients who do not understand English or Luganda, the two languages commonly used for communication at The Centre. They also noted that the consent forms are only available in English.

*Usually the biggest hindrance is language [. . .] But a patient with whom we speak English or Luganda we understand each other's communication, are usually open and they really want to* know. *(UCI 05)*

All respondents complained about the lack of privacy in the outpatient clinic. They felt that this prevents patients from communicating freely. This was attributed to the limited clinic space and big patient numbers attending the clinic.

*We would like to have more clinic space for the big numbers of patients, more spacious rooms to ensure full privacy [. . .] (UCI 06)*

## Recommendations for improving the consenting process for cancer care

Several suggestions for improving healthcare provider-patient communication were proposed. Lack of sufficient basic knowledge on cancer by the individuals who solicit informed consent was identified as one of the leading barriers to patients' comprehension of the information

disclosed during consenting process. Two respondents suggested that informed consent should be obtained by healthcare professionals who are knowledgeable and have good understanding of oncology. Respondents also recommended the training of healthcare professionals and equipping them with basic knowledge on oncology, and skills in effective communication, including the breaking of bad news.

> "*Health care workers should be trained to be more sensitive to patients concerns and be more responsive. We need to train health care providers on communication skill. How do you communicate to a patient? What is the best time, language, face you need to put on, the tone to communicate, the content? How do you check that you have communicated? We need to equip the people; we need to train them.* (UCI 09)*

Continuing medical education and training of health professionals in medical ethics was strongly recommended. Respondents also felt that patients should also be sensitised about the importance of informed consent in clinical practice.

> *Now the other aspect also is we need to do it from both ends, we need to get workers to appreciate the importance of informed consent as a protective measure because others take it lightly and from the perspective of patients, patients also need to appreciate that they need to consent. (UCI 09)*

Another suggestion was the developing of appropriate and detailed informed consent documents, and standard operating procedures to guide the consenting process. Respondents suggested that consent documents should be translated into the most spoken local languages.

> *Have different consent forms for different procedures not a general one. It makes a lot of sense when someone documents that is consenting for a specific treatment, whether it's a procedure whether it's an infusion, whether it is swallowing it makes a lot of sense. (UCI 05)*

Limited clinic space was identified as the main impediment to protection of privacy. Therefore, respondents recommended improvement of physical infrastructure to accommodate the large volume of patients and improve on patient privacy and confidentiality.

> *We would like to have more clinic space for the big numbers of patients, more spacious rooms to ensure full privacy* [...] (UCI 06)

Another issue that emerged was the role of patients in the decision-making process. One respondent recommended patient should be sensitized on their right to informed consent, and more involvement of patients in the decision-making process.

## Discussion

Overall, all respondents had a clear understanding of informed consent and what the consenting process entails. They appreciated the value of disclosing accurate information to patients during the informed consent process to facilitate informed decision making. They concurred that patients have a right to information, and their expectations and preferences should be considered in making treatment decisions. However, they noted that at times, it is difficult to fully respect patients' preferences. On the other hand, all respondents in our study expressed dissatisfaction with the consenting process at The Centre and highlighted major weaknesses in the consenting process. These findings are consistent with several studies in the literature that

have reported sub-optimal informed consent practices in clinical care in Uganda [21, 23–27]. A recent study at Uganda Cancer Institute reported that more than half of the participants had little knowledge about their disease and treatment and 41% were never counselled about the side effects of therapy [21]. This further reiterates the need to identify the challenges and barriers to optimize the informed consent process for cancer care in low resource settings.

The patient has the right to know and participate in choosing his/her therapy. However, without basic understanding of the disease, risks, benefits, and alternative treatment options, patients cannot meaningfully make informed choices. Respondents felt that doctors are best suited to disclose disease and treatment-related information since they are perceived to be knowledgeable and well-grounded in oncology. However, this duty is often delegated to nurses and other junior healthcare professionals who may not be knowledgeable. It is important to note that disclosure of the diagnosis and prognostic information is a challenge to healthcare professionals the world over [33–35]. Consenting to cancer therapy is not just a matter of signing a consent form after receiving a simple explanation. With complex treatments becoming increasingly available in low resourced settings, it is ever becoming more important that healthcare professionals clearly understand and appreciate the treatment options, potential harms, and therapeutic outcomes of cancer therapy. It is also important to consider the social and cultural context in which consent is obtained because it can impact the information that should be disclosed to the patient [36]. That is why it is important for healthcare professionals to have meaningful dialogue with patients and their surrogates during the consenting process for cancer care. This not only helps clinicians learn about patients' expectations and what they actually want to know about their disease and treatment, but also makes it easier for them to give appropriate explanations in case there is divergence in opinions and preferences.

Cancer care requires a multi-disciplinary approach involving various actors, including patients' families. Family members play an important role in providing care to patients with cancer and should therefore participate in making decisions on the nature of therapy. It is also important to have family members that clearly understand and accept the patient's health condition and prognosis. Such family members often have realistic expectations [9]. Evidence suggests that family involvement in decision making is associated with enhanced patient satisfaction and adherence to treatment [37]. Doctors in this study reported that oftentimes they convene meetings between the medical team and the patient's family to decide on how best to communicate potentially distressing information to the patient, especially for the very sick and terminally ill. Breaking bad news to cancer patients is never easy [38, 39]. In several parts of the world, physicians prefer to discuss with families and break bad news to family members and let them decide whether the patient should be informed [40–43]. However, this creates a value conflict between respecting the patients right to know and respecting the family's desire to protect the patient [40–43]. Disclosing the truth when breaking bad news is essential for informed decision-making for treatment. However, at times it is necessary to withhold the truth from the patient. Clinicians should assess the patient's readiness to receive sensitive information. Patients should first be psychologically prepared before communicating potentially distressing information.

Withholding the truth and delayed disclosure of information from cancer patients is relatively common globally [41, 42]. Disclosure that involves withholding or misrepresentation of information is referred to as "*fraudulent disclosure*" and undermines consent [36]. Fraudulent disclosure can lead to invalid consent whereby a physician illegitimately controls a patient's decision by controlling the information he/she discloses. For example, a physician might withhold information on drug adverse effects because he/she benevolently would not like to worry the patient. Despite the physician's good intentions, if the information he/she withholds is dispositive of the patient's decision to consent to treatment, he/she exercises illegitimate control

that invalidates consent [36]. On the other hand, there is the concept of "therapeutic privilege", where a physician can legitimately withhold information based on sound medical judgement, and disclosure of such information would potentially harm an emotionally drained or unstable patient [1]. Many people believe that disclosure of information on the nature and prognosis of cancer can result in emotional reactions that may impact negatively on treatment [41]. Withholding of information is therefore aimed at protecting patients, to give them hope, and encourage them to adhere to treatment [42]. However, contrary to the above argument, Giacomo et al. [44], contended that lack of information can cause uncertainty, anxiety and dissatisfaction and could be detrimental to the patient-physician relationship [45]. We therefore argue that withholding information should be based on sound medical judgement and not what the healthcare professional or family desire.

With the fast-paced medical and technological advancement, the informed consent process and thus patient education information is becoming ever more extensive and complex [46]. And that is the main reason why many clinicians are employing decisional aids to enhance comprehension and facilitate informed decision making in cancer care [47–49]. However, in this study we noted that much as IEC materials were available in the clinic, they were insufficient and healthcare professionals seldom use them. Some literature claims that patient IEC materials on cancer treatment are lengthy, difficult to understand, and often not suitable for laymen [50]. The IEC should therefore be tailored to the health literacy level of the public. In order to enhance comprehensibility, the vocabulary and language should be simplified to suit the target group [50].

## Strengths and limitations

This is the first study to investigate and report in-depth perspectives of healthcare professionals of the informed consent process in cancer clinical care in Uganda. Furthermore, all respondents were actively participating in patient education and the administration of informed consent at a premier cancer treatment institution, as such, their insights will significantly inform best practices for cancer care. The main weakness of the study was the recruitment of healthcare professionals from only one institution (The Centre). However, we believe their views are provide a glimpse of what happens at most hospitals in Uganda, since our findings resonate well with the literature.

Regarding research reflexivity, we were aware that when interviewing healthcare professionals, we tried to remain neutral, setting aside our own views and to listen from the respondents' perspective.

There were a few differences in the perspectives of the different health professionals. For confidentiality purposes, the professional roles of the respondents have been excluded in the presentation of results. Unlike nurses, doctors and the health educator called for meaningful dialogue and increased involvement of patients in consent discussions. They were also critical of the rudeness and unprofessional conduct of some of the personnel that solicit consent from patients. On the other hand, nurses and the health educator were critical of the delegation of duties to unqualified personnel. They asserted that doctors should devote adequate time to consent processes because their in-depth knowledge and understanding of cancer and cancer treatment.

## Conclusion

In summary, respondents appreciated the value of disclosing accurate information to patients to facilitate informed decision making. They all concurred that patients have a right to information, and their expectations and preferences should be considered during the consenting process. However, they noted that at times, it is difficult to fully respect patients' preferences.

Several challenges to optimal information disclosure and informed consent for cancer care at The Centre have been identified. Healthcare professionals in this study expressed discontent with the informed consent practices because of the several challenges they encounter during the course of their duties. Informed consent documentation is poor, there is lack of privacy, and some healthcare professionals have poor communication skills. Some healthcare professionals behave unprofessionally and are poor communicators, and this impacts on patients' mental health and motivation to contribute to meaningful dialogue.

Ideally, physicians are the ones supposed to solicit informed consent from patients, however, they usually delegate this responsibility people who may not be competent enough to communicate technical information.

These challenges were attributed to the lack of manpower coupled with a heavy patient load, time constraints, limited space, bad attitude towards patients, linguistic and ethnic barriers, and lack of policies and standard operating procedures on informed consent. Healthcare professional often liaise with patients' family members for assisted decision-making and determine which information should be disclosed to the patient without causing harm; especially the terminally ill. Therefore, whenever necessary, family members should be considered in making decisions during the informed consent process, particularly for life-threatening diseases like cancer.

## Best practices

We recommend significant improvement in informed consent practices in Uganda. There is a need for improved documentation through the development of detailed context-specific informed consent forms which should also be translated into the commonly spoken local language. Institutional policies and standard operating procedures should also be developed to guide patient health education and the consenting process.

One of the major challenges observed in this study was lack of privacy that was mainly attributed to the limited space in the out-patient clinics. We recommend physical infrastructural capacity strengthening to accommodate the large volume of patients and improve patient privacy and confidentiality. However, such a venture is very expensive and requires proper planning. We therefore implore UCI to consider this in their strategic planning.

There is a need for a less individualistic view of personal autonomy. Just like Ochieng et al. [51], we recommend interactive informed consent approaches that promote active patient involvement and bi-directional communication. A patient-centred approach to informed consent could help healthcare professionals see information disclosure as an opportunity to help patients cope and cooperate, rather than trying to avoid psychological distress that may already be present [43]. Such an approach could also help healthcare professionals understand the patient better, provide an opportunity to manage the patient's expectations, and address any misperceptions.

## Educational implications

We recommend continuing medical education and training of healthcare professional and equipping them with basic knowledge on the common cancers and their treatment. They should receive short course training in medical ethics with specific emphasis on patient autonomy and the informed consent process. These healthcare professionals should also undergo training in effective communication, counselling, and how to break bad news to cancer patients and their families. By so doing, we hope that this will enhance the communication skills of healthcare professionals and ultimately improve patients' comprehension during the informed consent process.

### Research agenda

Several studies in the literature have consistently documented major flaws in informed consent practices in clinical care in Uganda [23–27]. Most of these studies were conducted in tertiary and university teaching hospitals that may not be representative of the situation in the entire country. A multi-disciplinary nationwide needs assessment should be conducted to identify the current practices, facilitators, and barriers/challenges to optimal informed consent. We posit that this information will be valuable in establishing possible solutions and informing policy to optimize informed consent best practices in Uganda.

## Supporting information

**S1 Checklist. COREQ checklist.**
(DOCX)

**S1 File. Qualitative dataset.**
(DOCX)

**S2 File. Interview guide.**
(DOCX)

## Acknowledgments

We acknowledge Associate Professor Charles Ibingira who was a co-supervisor to RK. We would like to thank UCI management for granting us administrative clearance to conduct research at her premises. We also acknowledge all our research participants and UCI staff. Lastly, we acknowledge Adelline Twimukye for her role in data analysis. The authors would like to acknowledge all respondents.

## Author Contributions

**Conceptualization:** Rebecca Kampi, Erisa Sabakaki Mwaka.

**Data curation:** Rebecca Kampi, Clement Okello, Joseph Ochieng, Erisa Sabakaki Mwaka.

**Formal analysis:** Rebecca Kampi, Clement Okello, Joseph Ochieng, Erisa Sabakaki Mwaka.

**Investigation:** Rebecca Kampi, Clement Okello, Erisa Sabakaki Mwaka.

**Methodology:** Rebecca Kampi, Erisa Sabakaki Mwaka.

**Project administration:** Rebecca Kampi.

**Resources:** Rebecca Kampi.

**Supervision:** Joseph Ochieng, Erisa Sabakaki Mwaka.

**Validation:** Rebecca Kampi, Clement Okello, Joseph Ochieng, Erisa Sabakaki Mwaka.

**Visualization:** Rebecca Kampi, Erisa Sabakaki Mwaka.

**Writing – original draft:** Rebecca Kampi.

**Writing – review & editing:** Rebecca Kampi, Clement Okello, Joseph Ochieng, Erisa Sabakaki Mwaka.

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
