## [Decision Letter · Decision Letter 0]

22 Nov 2023

PONE-D-23-26045Information disclosure during the consenting process in cancer clinical care: perspectives of healthcare professionals at Uganda Cancer InstitutePLOS ONE

Dear Dr. Mwaka,

Thank you for submitting your manuscript to PLOS ONE. After careful consideration, we feel that it has merit but does not fully meet PLOS ONE’s publication criteria as it currently stands. Therefore, we invite you to submit a revised version of the manuscript that addresses the points raised during the review process.

We look forward to receiving your revised manuscript.

Kind regards,

Luca Valera

Academic Editor

PLOS ONE

A clean copy of the edited manuscript (uploaded as the new *manuscript* file).

“This study was supported by the Fogarty International Center of the National Institutes of Health under Award Number R25TW009730 that supported RK’s Master of Health Sciences in Bioethics program at Makerere University. The content is solely the responsibility of the authors and does not necessarily represent the official views of the National Institutes of Health. The authors would like to acknowledge all respondents.”

“We acknowledge Associate Professor Charles Ibingira who was a co-supervisor to RK. We would like to thank UCI management for granting us administrative clearance to conduct research at her premises. We also acknowledge all our research participants and UCI staff. Lastly, we acknowledge Adelline Twimukye for her role in data analysis.

This study was supported by the Fogarty International Center of the National Institutes of Health under Award Number R25TW009730. The content is solely the responsibility of the authors and does not necessarily represent the official views of the National Institutes of Health. The authors would like to acknowledge all respondents.”

“This study was supported by the Fogarty International Center of the National Institutes of Health under Award Number R25TW009730 that supported RK’s Master of Health Sciences in Bioethics program at Makerere University. The content is solely the responsibility of the authors and does not necessarily represent the official views of the National Institutes of Health. The authors would like to acknowledge all respondents.”

Reviewers' comments:

Reviewer's Responses to Questions

**Comments to the Author**

1. Is the manuscript technically sound, and do the data support the conclusions?

Reviewer #1: Partly

Reviewer #2: Partly

2. Has the statistical analysis been performed appropriately and rigorously? 

Reviewer #1: N/A

Reviewer #2: N/A

3. Have the authors made all data underlying the findings in their manuscript fully available?

Reviewer #1: Yes

Reviewer #2: Yes

4. Is the manuscript presented in an intelligible fashion and written in standard English?

Reviewer #1: No

Reviewer #2: Yes

5. Review Comments to the Author

Reviewer #1: The paper presents an interesting research based in Uganda about healthcare professionals’ perception of information disclosure to patients in the sensitive context of cancer care.

My premise is that I strongly believe this kind of qualitative research is fruitful since it gives the opportunity to the main actors of a specific context to express their opinion. However, it must be rigorous, and the methodology must be solid and well described: otherwise, the high risk is for qualitative research to be biased and provide opinions instead of precious findings.

I chose to give the authors the chance to show me that the research is based on a solid methodology, that unfortunately I cannot see in the article. Here some specific points I find weak:

- In the Introduction I read “one way of respecting a person’s autonomy is through obtaining informed consent”. This sentence shows me a general problem about this article: instead, I would say that respecting one’s autonomy entails giving them the possibility to give their consent, that must be informed. Put in the authors’ way, this sensitive topic is quite shallow. I trust that the authors know what they are talking about, but they did not manage to well explain it in the article. Also, there is no literature about autonomy nor informed consent: adding it could help the authors’ digging into the main topic of their research in a more profound way, that consequently helps the reader to understand the overall picture.

- Although every informed consent should be considered essential in persons’ care, the authors want to be sure that their readers understand that different fields of care are on different levels. Comparing the informed consent process for dental care to cancer care it is confusing and does not give to the sensitive context they are exploring the attention it deserves. Again, no literature about the sensitive context of cancer care is provided.

- What’s the Charter (reference 8)? It is not obvious for a reader that is not from Uganda. No specific understanding of Uganda’s context is provided – except for a brief description of the hospital where they conducted their research. I think that it is useful to describe every context in its uniqueness, and especially such a context: it would give to the research more credibility and show its importance. From authors’ words, “the required services are not available at the primary health unit where patients are first attended”: what does this mean? That is not common and probably pertains to a delicate context that is not explained. The “linguistic challenge” that such a context must face are not obvious either.

- “Materials and methods” section is quite weak. I would prefer a different structure to give a solid base to the research. I cannot find different important elements; for example, why only 10 participants? It’s quite a small number. Even more important, why only healthcare professionals and not patients, that are the main subjects of the topic? I would suggest providing a table about interviewees, with variables that the authors seemed to collect (profession, gender, age…). Also, it would be interesting to provide a table with the questions they used in the interviews; and were the interviews structured, semi-structure…? What do the authors mean when they say that “the interview guide was piloted on three cancer patients”? It is not a standard process considering that they aimed to interview healthcare professionals and not patients, so they at least need to provide a justification for that. The authors declare that translations from Luganda to English were made: what is the method implied? Was the translation provided by a mother tongue? Or a software? That is such an important point, because we well know that translations may convey different meanings compared to the original language. The authors declare that “some transcripts were returned to interviewees for member checking”: what does this mean? Is this related to the coding? This point also needs a justification. A description of the interviewers is provided at the end of the section dedicated to data management and analysis, and I do not think it is the right place. Also, the authors disclose their biases in conducting this research, but the way they do that (“it was difficult for us to be totally objective”) does not strengthen the article; instead, it could be seen as unprofessional and compromise the overall research, because even though it is true that researchers are not objective, their methodology should be in order to guarantee reliable results.

- The authors report that “some respondents […] withhold or delay the disclosure of vital information”: this is such an interesting and sensitive point that must be further explored – otherwise, it could be read as a justification for this behavior that is not even legally permitted. Similarly, when the authors say that some healthcare professionals “hold discussions with the patient’s family if the patient is not competent”: what do they mean with “competent”? Can ever a patient be competent enough? Again, “patients are increasingly agitating for more active involvement” is an ambiguous sentence that needs to be further explored.

- References 28 and 32 are not written in the correct way.

- The “Discussion” section often repeats concepts already explored without digging more. As already said, sentences like “at times it is necessary to withhold the truth from the patient” seem to me quite dangerous if not well-contextualized. It is quite annoying for the reader that some sentences are repeated in different places without further analysis.

- I suggest a revision of English: there are subjects missing and many sentences are just copied and pasted.

In my opinion, the subject is interesting and I would be glad to read this article written in a more solid way; I expect the authors to show the robustness of their methodology based on the comments above.

Reviewer #2: See uploaded document that includes all comments to the author as the formatting became distorted when uploaded to this box. Please note am also typing here to meet character limit so I can upload document successfully and proceed with submission of this review to PLOS One.

6. PLOS authors have the option to publish the peer review history of their article (what does this mean?). If published, this will include your full peer review and any attached files.

Reviewer #1: **Yes: **Monica Consolandi

Reviewer #2: No

---

## [Author Response · Author response to Decision Letter 0]

6 Jan 2024

Reviewer 1 

Comment Response Page number

In the Introduction I read “one way of respecting a person’s autonomy is through obtaining informed consent”. This sentence shows me a general problem about this article: instead, I would say that respecting one’s autonomy entails giving them the possibility to give their consent, that must be informed. Put in the authors’ way, this sensitive topic is quite shallow. I trust that the authors know what they are talking about, but they did not manage to well explain it in the article. Also, there is no literature about autonomy nor informed consent: adding it could help the authors’ digging into the main topic of their research in a more profound way, that consequently helps the reader to understand the overall picture. We would like to that the reviewer for this important observation. The entire introduction has been re-written. Informed consent has been defined and the other factors that influence the informed consent process have been added. Information has been added on the Ugandan context. Pages 4- 6

Although every informed consent should be considered essential in persons’ care, the authors want to be sure that their readers understand that different fields of care are on different levels. Comparing the informed consent process for dental care to cancer care it is confusing and does not give to the sensitive context they are exploring the attention it deserves. Again, no literature about the sensitive context of cancer care is provided.

- What’s the Charter (reference 8)? It is not obvious for a reader that is not from Uganda. No specific understanding of Uganda’s context is provided – except for a brief description of the hospital where they conducted their research. I think that it is useful to describe every context in its uniqueness, and especially such a context: it would give to the research more credibility and show its importance. From authors’ words, “the required services are not available at the primary health unit where patients are first attended”: what does this mean? That is not common and probably pertains to a delicate context that is not explained. The “linguistic challenge” that such a context must face are not obvious either. The entire introduction has been re-written. Informed consent has been defined and the other factors that influence the informed consent process have been added. Information has been added on the Ugandan context. All the statements in question have been deleted. Pages 4- 6

“Materials and methods” section is quite weak. I would prefer a different structure to give a solid base to the research. I cannot find different important elements; for example, why only 10 participants? It’s quite a small number. Even more important, why only healthcare professionals and not patients, that are the main subjects of the topic? Thanks for the comment. This study focused on healthcare professionals because several studies have reported challenges to the informed consent process in clinical care in Uganda, with healthcare professionals being the main culprits. That is why we set out to examine the opinions of healthcare professionals as opposed to patients. This information has been included under “Participants”.

We conducted on 10 interviews because data saturation was achieved after the 10th interview. All respondents were all from the same institution. This has been included on pages 7- 8 

I would suggest providing a table about interviewees, with variables that the authors seemed to collect (profession, gender, age…). 

 A table showing the sex and profession of the respondents, in general terms has been added on page 10. This has been done to protect the confidentiality of the respondents since the number is small and all are from the same institution. Pages 10

Also, it would be interesting to provide a table with the questions they used in the interviews; and were the interviews structured, semi-structure…? We used a semi-structured interview guide (Uploaded as a supplementary file).

What do the authors mean when they say that “the interview guide was piloted on three cancer patients”? It is not a standard process considering that they aimed to interview healthcare professionals and not patients, so they at least need to provide a justification for that. We sincerely apologize for this error. The tool was pre-tested on three healthcare professionals offering out-patient care at another hospital. 

The authors declare that translations from Luganda to English were made: what is the method implied? Was the translation provided by a mother tongue? Or a software? That is such an important point, because we well know that translations may convey different meanings compared to the original language We apologise for this error, all interviews were conducted in English. 

The authors declare that “some transcripts were returned to interviewees for member checking”: what does this mean? Is this related to the coding? This point also needs a justification. Member checking refers to the return of transcript to respondents to check for accuracy of transcription. This has been changed. Page 9

A description of the interviewers is provided at the end of the section dedicated to data management and analysis, and I do not think it is the right place. Also, the authors disclose their biases in conducting this research, but the way they do that (“it was difficult for us to be totally objective”) does not strengthen the article; instead, it could be seen as unprofessional and compromise the overall research, because even though it is true that researchers are not objective, their methodology should be in order to guarantee reliable results. We would like to thank the reviewer for this observation. We included that as a statement on reflexivity based on the COREQ requirement for reporting qualitative research. We have revised the statement and transferred it to the study limitations. Pages 25

The authors report that “some respondents […] withhold or delay the disclosure of vital information”: this is such an interesting and sensitive point that must be further explored – otherwise, it could be read as a justification for this behavior that is not even legally permitted. Withholding or delayed disclosure of information is commonly practiced by health professional globally, but it must be based on sound medical judgement. It is also documented in ethics literature and is often referred to as “therapeutic privilege”. This issue is discussed at length in the discussion and references have been cited. Pages 23- 24

Similarly, when the authors say that some healthcare professionals “hold discussions with the patient’s family if the patient is not competent”: what do they mean with “competent”? Can ever a patient be competent enough? Again, “patients are increasingly agitating for more active involvement” is an ambiguous sentence that needs to be further explored. These statements have been revised 

References 28 and 32 are not written in the correct way. Both references have been deleted 

The “Discussion” section often repeats concepts already explored without digging more. As already said, sentences like “at times it is necessary to withhold the truth from the patient” seem to me quite dangerous if not well-contextualized. It is quite annoying for the reader that some sentences are repeated in different places without further analysis. This has been revised and clarified in the discussion Pages 23- 24

Reviewer 2 

Need to clarify what is meant by ‘informed consent’ - this term has multiple interpretations and conceptualizations, depending on the context. It can mean a specific document related to a specific event (e.g., surgery or procedure) or it can relate to more general discussions between providers and patients regarding prognosis, treatment goals, and disclosure of diagnosis. In the paper, multiple aspects of ‘informed consent’ are discussed and what is meant by ‘information disclosure during the consenting process’ is unclear. It also seems based on the interview data participants interpreted this concept very differently as well. 

 We are very grateful for this comment. We have provided more literature on informed consent in the introduction. Pages 4- 6

Inappropriate qualitative methodology and concern regarding sampling strategy. Sample size concern

How was data saturation achieved given n=10 of such diverse participant perspectives and roles? We conducted on 10 interviews because data saturation was achieved after the 10th interview. All respondents were purposively recruited from one institution. This has been included on page 8 

Abstract:

- Needs more detail and organization, especially regarding methods.

o Introduction – need to specify study took place in Uganda.

o Methods – more detail is needed (e.g, type of healthcare providers recruited; how were key informants selected; within same institution; what qualitative approach was used? How were interviews conducted?

Results – should be organized by key themes; barriers are referenced but not listed/described The abstract has been revised. 2

Introduction:

- Need to place study within global context and acknowledge differing cultural norms related to ‘informed consent’

- Need to clarify how the proposed study address the gap in the existing literature

- Similarly, need to provide more citations supporting the need to look at informed consent processes in cancer care specifically.

 We would like to that the reviewer for this important observation. The entire introduction has been re-written. Informed consent has been defined and the other factors that influence the informed consent process have been added. Information has been added on the Ugandan context. 

Methods:

- Sections disjointed and not well organized

- More detail regarding how key informants were selected and rationale for sampling strategy is needed

Potential sampling issue – my experience in global health is that nurses are very rarely/if ever involved in ‘obtaining informed consent’ – although this is the eligibility criteria described under “Participants.” See comment below about being clear what is meant by ‘informed consent.’ The material and methods’ section has been revised. Unnecessary information has been deleted, and some transferred to the study limitations.

We totally agree with the reviewer. Ideally it is the physician who should solicit informed consent from patients. However, the few studies from Uganda have reported non-compliance with internationally accepted standards regarding informed consent for clinical care. These studies have shown that it is the nurses and other junior staff that solicit informed consent, including for surgical procedures. (Kituuka: doi.org/10.1186/s12873-023-00856-0, Ochieng: http://www.biomedcentral.com/1472-6939/15/40). In another study only 5% of dental practitioners obtained written informed consent for invasive dental procedures (Nono: https://doi.org/10.1186/s12903-022-02550-2).

The situation is not any different at the Uganda Cancer Institute, it is the junior staff that solicit consent, and that is why we selected the people who actively participate in the consenting process. 

 6- 9

Include more detail about type of institution and rationale for study site in Methods – did not de-identify study site – seems problematic given small sample size More detail has been added to the description of the study site. The study site has also been de-identified to protect the confidentiality of the study participants. Pages 7

Authors state they used a phenomenological approach to their qualitative inquiry, but this is not supported by their reported data collection and analysis methods. Phenomenology is a highly specialized and complex type of qualitative inquiry that deeply explores the ‘lived experience’ of human existence (such as grief), with in-depth interviews (often repeated interviews with same participants) grounded in the philosophical orientation of phenomenology and requires a high level of abstraction in data analysis. What is described in this paper is consistent with a qualitative descriptive approach, where the goal is to stay close to the data to answer more concrete and specific questions that have been identified a priori (see Sandelowski, PMID: 10940958) We would like to thank the reviewer for this insightful comment. We would also like to thank him for the article. We have revised the approach to a “Qualitative descriptive” approach. 

Interviews were all in English, but some were translated from local language? We apologise for this error, all interviews were conducted in English.

141-142 – Why was the interview guide pilot tested on cancer patients? We sincerely apologise for this error. The tool was pre-tested on three healthcare professionals offering out-patient care at another hospital. Pages 8

Data analysis – where is prior analytic framework? (referenced in lines 162-164) The pre-determined themes are included in the interview guide that has been uploaded as a supplementary file.

166-167 – quotes should inform themes not vice versa We indicated that the themes emerged from the text.

169-180 – feels out of place with unnecessary detail; could add a brief statement regarding reflexivity to Discussion or perhaps to Limitations Details on reflexivity have been transferred to the limitations. Pages 25

Results:

Would help to describe participants and their roles more clearly to help contextualize the themes – e.g., what is meant by medical officer? What is the difference between nurse and nurse counselor? Is the palliative care specialist a physician or a nurse? - A medical officer is a general practitioner, and a nurse counsellor is a nurse who specializes in counselling. The palliative care specialist is a medical consultant. In view of the comment from reviewer 2, we have included a table with only sex and occupation. We have also decided to use “Doctor” for the oncologists, medical officer and palliative care specialist to protect their confidentiality, since the sample size was small. We have also refrained from indicating their professional role against the quotes.

 Pages 10

Table 1 would be strengthened by including supporting quotes to justify how themes were developed. Thanks for the suggestion, we have added a few quotes to Table 2

Overall, Results section could be more concise and more clearly organized (some information is repeated – e.g., suggestions are repeated again under ‘barriers’)

 The results section has been improved and redundant quotes deleted. 

Consider a table that summarizes recommendations discussed by participants in interviews The recommendations are included in Table 2 under the theme The recommendations are included in Table 2 under Theme 5: Recommendations for improving the consenting process for cancer care. Pages 11

439-443 – repeated quote. Thanks for this observation, the quote has been 

Need to clarify the role the family plays in receiving the information and providing informed consent (e.g., 224 – 225)

 Details on the role of the family in informed consent have been included in the discussion. Pages 22- 23

Discussion:

- Some information in Discussion is repeated information from Results

- Discussion should focus on how findings differ or support prior work; how findings should be interpreted; how fills a gap in the literature (as it reads, it’s unclear the main contributions or innovation the paper makes to the current literature)

- Differences in responses by discipline/background could be significantly influenced by scope of practice, perceived roles, issues related to power/control, cultural norms, structural barriers related to communication – and critically how participant understood the concept of ‘informed consent.’ These factors are not adequately addressed and need more attention.

555 – 557 - “It is for this reason that some authors believe that physician led informed consent discussions are often ineffective because of the heavy workload and mechanistic way consent is obtained [41].” What do the authors mean by “some authors”? Why is there an implied discrepancy between authors The discussion has been significantly improved and information and quotes deleted.

This has been deleted. 

Best practices:

- Would cut this section as does not feel relevant to this paper; reads as section of dissertation inserted into manuscript. 

 This section highlights some of the recommendations from our study. It is now more concise; the irrelevant information has been deleted. Pages 27

Minor point 

- Term ‘opined’ is overused throughout paper and is a bit distracting

 This has been revised.

---

## [Decision Letter · Decision Letter 1]

19 Mar 2024

Informed consent in cancer clinical care: perspectives of healthcare professionals on information disclosure at a tertiary institution in Uganda

PONE-D-23-26045R1

Dear Dr. Sabakaki Mwaka

We’re pleased to inform you that your manuscript has been judged scientifically suitable for publication and will be formally accepted for publication once it meets all outstanding technical requirements.

An invoice for payment will follow shortly after the formal acceptance. To ensure an efficient process, please log into Editorial Manager at Editorial Manager® , click the 'Update My Information' link at the top of the page, and double check that your user information is up-to-date. If you have any billing related questions, please contact our Author Billing department directly at authorbilling@plos.org.

Kind regards,

Luca Valera

Academic Editor

PLOS ONE

Additional Editor Comments (optional):

Reviewers' comments:

Reviewer's Responses to Questions

**Comments to the Author**

1. If the authors have adequately addressed your comments raised in a previous round of review and you feel that this manuscript is now acceptable for publication, you may indicate that here to bypass the “Comments to the Author” section, enter your conflict of interest statement in the “Confidential to Editor” section, and submit your "Accept" recommendation.

Reviewer #1: All comments have been addressed

2. Is the manuscript technically sound, and do the data support the conclusions?

Reviewer #1: Yes

3. Has the statistical analysis been performed appropriately and rigorously? 

Reviewer #1: N/A

4. Have the authors made all data underlying the findings in their manuscript fully available?

Reviewer #1: Yes

5. Is the manuscript presented in an intelligible fashion and written in standard English?

Reviewer #1: Yes

6. Review Comments to the Author

Reviewer #1: "To the best of the authors’ knowledge, this is the first article to document informed consent practices in cancer care in Uganda."  I would put this line 101 before “this study therefore…”

I'm glad to see that the authors addressed all the comments and suggestions; the article sounds definitely better. I'm also happy to see that this kind of qualitative research is being pursued, providing PlosOne readers with the opportunity to better understand a particular context, such as the Ugandan one.

7. PLOS authors have the option to publish the peer review history of their article (what does this mean?). If published, this will include your full peer review and any attached files.

Reviewer #1: **Yes: **Monica Consolandi

---

## [Editor Report · Acceptance letter]

22 Mar 2024

PONE-D-23-26045R1 

PLOS ONE

Dear Dr. Mwaka, 

I'm pleased to inform you that your manuscript has been deemed suitable for publication in PLOS ONE. Congratulations! Your manuscript is now being handed over to our production team.

Kind regards, 

on behalf of

Dr. Luca Valera 

Academic Editor

PLOS ONE